# The Development and Application of Opto-Chemical Tools in the Zebrafish

**DOI:** 10.3390/molecules27196231

**Published:** 2022-09-22

**Authors:** Zhiping Feng, Bertrand Ducos, Pierluigi Scerbo, Isabelle Aujard, Ludovic Jullien, David Bensimon

**Affiliations:** 1Department of Chemical and Systems Biology, Stanford University, Stanford, CA 94305, USA; 2Laboratoire de Physique de l’Ecole Normale Supérieure, Paris Sciences Letters University, Sorbonne Université, Université de Paris, Centre National de la Recherche Scientifique, 24 Rue Lhomond, 75005 Paris, France; 3High Throughput qPCR Core Facility, Ecole Normale Supérieure, Paris Sciences Letters University, 46 Rue d’Ulm, 75005 Paris, France; 4Inovarion, 75005 Paris, France; 5Laboratoire PASTEUR, Département de Chimie, Ecole Normale Supérieure, Paris Sciences Letters University, Sorbonne Université, Centre National de la Recherche Scientifique, 24 Rue Lhomond, 75005 Paris, France; 6Department of Chemistry and Biochemistry, University of California, Los Angeles, CA 90095, USA

**Keywords:** zebrafish, opto-chemical tools, photo-activable molecules, single-cell physiology

## Abstract

The zebrafish is one of the most widely adopted animal models in both basic and translational research. This popularity of the zebrafish results from several advantages such as a high degree of similarity to the human genome, the ease of genetic and chemical perturbations, external fertilization with high fecundity, transparent and fast-developing embryos, and relatively low cost-effective maintenance. In particular, body translucency is a unique feature of zebrafish that is not adequately obtained with other vertebrate organisms. The animal’s distinctive optical clarity and small size therefore make it a successful model for optical modulation and observation. Furthermore, the convenience of microinjection and high embryonic permeability readily allow for efficient delivery of large and small molecules into live animals. Finally, the numerous number of siblings obtained from a single pair of animals offers large replicates and improved statistical analysis of the results. In this review, we describe the development of opto-chemical tools based on various strategies that control biological activities with unprecedented spatiotemporal resolution. We also discuss the reported applications of these tools in zebrafish and highlight the current challenges and future possibilities of opto-chemical approaches, particularly at the single cell level.

## 1. Introduction

The regulation of biological processes in an organism relies on complex mechanisms that often require precise temporal and spatial control [1,2,3]. Consequently, being able to dissect and deconstruct these delicate processes in living systems is critical to understanding both normal and pathological physiology, notably at the cellular and molecular level. Over the past decades, tremendous progress has been made in designing various approaches to study physiological processes, utilizing a myriad of model organisms [4,5,6]. As one of the widely adopted vertebrate models, the zebrafish (*Danio rerio*) has been successfully used in the studies of development, behavioral genetics, human diseases, and drug discovery [7,8,9,10,11,12,13,14]. Some attractive attributes of zebrafish include ex vivo fertilization, high fecundity, rapid embryogenesis and growth, the optical clarity of embryos, ease of genetic manipulation, and cost-effective laboratory maintenance. In particular, embryonic transparency and permeability makes zebrafish an outstanding model to be coupled with optogenetics and opto-chemical tools. In this review, we highlight the development of opto-chemical tools that harness light-activatable molecules to control biological activities with high precision and describe their applications in zebrafish.

## 2. An Overview of Established Opto-Chemical Tools

To study biological processes in living systems, it is essential to have tools that can specifically modulate the targets of interest. Traditional approaches typically involve the treatment of target-specific modulators (e.g., agonists and antagonists) or genetic manipulations (e.g., gene knockdown, knockout, and knock-in). These conventional approaches are extremely valuable and have remarkably shaped our understanding of biology at the molecular and cellular level. However, a major challenge is to examine the activity of biological targets in space and time as they naturally act. Achieving the control of molecules at such scales requires fast and specific responding elements. As light inherently possesses exceptional spatiotemporal precision, photo-responsive molecules are great candidates for the modulation of biological activities with high spatiotemporal resolution. This review focuses on opto-chemical tools that combine chemistry with light to modulate activities of biomolecules. Optogenetic approaches relying on the genetic engineering of light-sensitive proteins have been extensively reviewed elsewhere [15,16,17,18,19,20]. The two most common opto-chemical strategies are photo-induced conformational changes (Figure 1A) and light-induced uncaging which typically implies chemical caging of small molecules (Figure 1B), oligonucleotides (Figure 1C), and peptides and proteins (Figure 1D).

### 2.1. Photo-Induced Conformational Change

Essential types of natural biological processes such as vision and phototropism rely on photo-induced conformational changes. During these processes, chromophores switch between two or more isomeric forms and subsequently alter the activities of their associated proteins. The reactions usually occur in a reversible manner to produce many rounds of inactive/active states, and for that reason, these photosensors are commonly called photoswitches [21]. Photoswitches can be generally categorized into two groups: photoreceptor derived, and non-photoreceptor based.

#### 2.1.1. Photoreceptor-Derived Photoswitches

The most convenient opto-chemical tools exist in nature and do not require further chemical modifications. These natural molecules are mostly found in photoreceptors of various organisms and typically change conformations upon light illumination. For example, photoreceptors in animals and some microbes rely on the seven-transmembrane-domain proteins called opsins [22]. Opsins use the retinal as a chromophore which isomerizes from the *cis* to *all-trans* form when exposed to light (Figure 2A), and the *all-trans* retinal is subsequently converted back to the *cis* retinal by a series of enzymes in the retina [23]. Another major chromophore harnessed by a host of bacteria and plants is the flavin group which includes riboflavin, flavin mononucleotide (FMN) and flavin adenine dinucleotide (FAD). Natural photoreceptors that require flavin groups include the light-oxygen-voltage sensing domain (LOV), sensors of blue-light using FAD (BLUF), and cryptochrome (CRY) [24,25,26]. Flavins absorb light and covalently bind to the cysteine (Cys) at the LOV active site of the photoreceptor to induce a conformational change (Figure 2B). Similar acting mechanisms apply to other natural chromophores such as tryptophan antennas from Arabidopsis UV-B resistance receptor [27,28,29] and bilin analogs from phytochromes [30,31].

#### 2.1.2. Non-Photoreceptor Based Photoswitches

Some biologically active molecules that are not coupled with natural photoreceptors in particular can also alter their activities upon light illumination. For example, when irradiated by UV light, retinoic acid, a critical signaling molecule during vertebrate development, isomerizes between *9-cis*, *13-cis* and *all-trans* forms which possess distinct biological activities [32,33,34]. Many naturally occurring dyes such as carbocyanine [35], rhodamine [36], and azobenzene [37] compounds are photosensitive and shift their fluorescence spectra after undergoing conformational changes. Some fluorescent proteins such as Dronpa and Padron can also respond to light activation in a reversible manner [38] while other photoactivatable fluorescent proteins irreversibly convert fluorescence such as Kaede, Dendra2, and mEos [39]. In the laboratory, synthetic chemistry has modified this natural toolkit to render photocontrol much more dynamic and versatile. A good example is azobenzene and its derivatives which are widely employed to achieve activity control via photo-induced conformational changes. Responding to both light and temperature, azobenzene and its derivatives can isomerize between the *Z* and *E* conformations (Figure 2C). By choosing different substituents on the benzene ring, the physical properties of azobenzene compounds such as solubility and excitation wavelength can vary greatly [40,41,42], facilitating their applications to diverse needs. Numerous azobenzene-based photo-switchable components have since been engineered and applied to study various biological processes both in vitro and in vivo [43,44,45,46,47,48,49,50]. Besides azobenzene-based photoswitches, other light-sensitive ligands [51] such as stilbene [52,53], hemithioindigo [54,55], spiropyran [56,57], and diarylethene-derived [58] molecules have also been explored as synthetic photoswitches.

### 2.2. Light-Induced Uncaging

Photoswitches based on conformational change are valuable opto-chemical tools, however, the vast majority of biological processes are not conformationally regulated by light. A more powerful and generic opto-chemical tool is light-induced uncaging in which a biomolecule of interest is covalently linked to a light-responsive protecting (caging) group. Light irradiation initiates a photolytic reaction with subsequent release of the biomolecule to function normally. In principle, photo-uncaging allows any molecules, small or large, to be chemically modified to become photoactivatable. The critical component of photo-uncaging is the caging group. The ideal caging group needs to be evaluated on several criteria: (1) it should inactivate the caged biomolecule without producing any secondary activity or toxicity; (2) the caged biomolecule should be stable outside and inside the biological system unless irradiated by light of the proper wavelength; (3) the light used should not be deleterious to the biological system and the photolysis reaction should be fast and high yielding; (4) any by-products from photo-uncaging should also be non-detrimental and preferably inert in the biological system. A variety of caging groups have been developed that satisfy these criteria fully or partially, and some of the most successful and prevalent caging groups (Table 1) include the 2-nitrobenzyl, coumarin, 7-nitroindolinyl and BODIPY derivatives [59,60,61,62]. Other caging groups have also been synthesized and tested in various contexts [63].

#### 2.2.1. Photocaged Small-Molecule Actuators and Probes

Small-molecule actuators and probes are essential tools for perturbing and investigating biological processes. Photo-uncaging of bioactive molecules provides additional control at the spatial and temporal level. The concept of small-molecule photo-uncaging dates back to the 1970s when cyclic adenosine monophosphate (cAMP) [64] and adenosine triphosphate (ATP) [65] were first photocaged. Since then, a wide range of small molecules including inhibitors, agonists, metabolites, and probes have been caged and successfully photo-activated in various biological systems [59,60,61,63,66,67,68,69,70,71]. Neurotransmitters were among the first biomolecules to be photo-uncaged to precisely control neuronal activities. By adding an ⍺-(4,5-dimethoxy-2-nitrobenzyl) group to glutamate, the caged molecule can be locally released via UV illumination [72]. Improved photo-uncaging of glutamate [73,74,75] as well as other neural signaling molecules such as γ-aminobutyric acid (GABA) [76,77] and calcium (Ca^2+^) [78,79] has also been achieved. In other cases, physiological metabolites such as inositol [80] and retinoic acid [81] can be photocaged to obtain spatiotemporal activation while photocaged dyes [82,83] have also been synthesized to facilitate biological labeling and imaging. Notably, small-molecule ligands that regulate signaling pathways or gene transcription have been broadly and fruitfully caged to render photocontrol. For example, rapamycin was caged to optically control the heterodimerization of the FK506 binding protein (FKBP) and FKBP-rapamycin binding protein (FRB) [84]. Caged doxycycline [85,86] and tamoxifen [87,88] were synthesized to activate gene expression with light. To improve photostability, caged 4-hydroxy-cyclofen (Cyc) [89,90,91,92] has also been developed to optimize its use in most physiological conditions. Optical control of gene expression with caged small molecules directly against mRNA is scarce, but recently, synthetic 5′ cap analogues with photo-cleavable groups have been developed [93]. UV irradiation efficiently releases the 5′ cap to interact with the eukaryotic translation initiation factor 4E (eIF4E), prompting the start of translation.

While photo-sensitive small molecules are mostly engineered towards their photo-activation, photo-deactivation approaches are less common. One method for photo-deactivation is chromophore-assisted light inactivation (CALI) in which engineered proteins and light-sensitive dye molecules produce reactive oxygen species (ROS) to deactivate biological systems upon light absorption [94,95]. However, due to the many limitations of the technique such as inconvenient cellular delivery and off-target ROS toxicity, CALI has not been broadly applied, especially in vivo. Another photo-deactivation approach with caged small molecules can be achieved through protein degradation. For example, a photoactivatable auxin has been shown to act as a photoactivatable inducer of protein degradation in mammalian cells in which transposing components of the plant auxin-dependent degradation pathway were genetically introduced [96]. Small-molecule degraders such as proteolysis-targeting chimeras (PROTACs) have also been caged to recruit the E3 ligase to the target protein upon light activation, resulting in ubiquitination and protein degradation [97,98]. Alternatively, PROTACs have also been designed as photoswitches that can be reversibly activated at different wavelengths [99,100,101].

Caging of small molecules is by far the most widely used photocaging technique due to the relatively straightforward yet highly effective design and synthesis. They are also ideal molecular tools for zebrafish study as most small molecules can freely diffuse in embryonic tissues and are readily accessible to light illumination.

#### 2.2.2. Caged Oligonucleotides

As many biological programs are initiated at the genetic level in cells, the ability to directly control DNA or RNA provides an additional layer of precision in studying biological processes. The idea of using nucleic acids as tools to control biological activity was realized soon after the “central dogma” was first proposed by Francis Crick in the late 1950s. However, developing light-activable oligonucleotides is challenging due to the large size and structural complexity [102]. Sub-optimal caging of oligonucleotides leads to either activity leakage while caged or insufficient activity upon photo-uncaging. Nevertheless, there have been great and promising developments in the synthesis of photocaged oligonucleotides in the past decades.

The first working example of photocaged oligonucleotide was obtained by adding several hundred 1-(4,5-dimethoxy-2-nitrophenyl) diazoethane groups to plasmids coding for luciferase and green fluorescent protein (GFP) [103]. Similarly, photo-uncaging of mRNA has also been reported to control the expression of *GFP* and *Engrailed2a*, a gene involved in zebrafish eye and brain development [104]. Efforts have been also made to cage small interfering RNA (siRNA). SiRNA is a class of double-stranded RNA of 20–24 base-pair nucleotides, and it interferes with gene expression by specifically targeting and degrading its complementary mRNA. Caged siRNAs were synthesized by adding photo-activable groups to the phosphate backbone [105], the 5′ terminal phosphate of the antisense strand [106] or the central region [107]. Further optimization of photocaged siRNAs that used new photolabile groups to induce tighter control have also been reported [108,109,110]. In zebrafish studies, the most common anti-sense knockdown tools use morpholino oligonucleotides (MOs) [111,112]. Instead of ribose rings, MOs contain morpholino rings and are neutrally charged by replacing the phosphate backbone with phosphorodiamidates [113]. MOs do not bind to the RNA-induced silencing complex (RISC). Consequently, MOs can effectively inhibit translation without inducing RNA degradation. Caged MOs were first developed by tethering complementary oligomers through photolabile linkers and were tested in zebrafish to silence gene expression of chordin and no tail (Ntl) in zebrafish embryos [114,115]. To avoid the use of multiple caging groups, MOs have also been circularized with a photocleavable linker joining the 3′-amine and 5′-carboxylic acid ends of linear oligonucleotides [116,117]. While most caged MOs (cMOs) are activated by UV light, various photolabile caging groups allow for non-UV photolysis and enable sequential control through wavelength-selective illumination [118]. MOs and cMOs have been widely used for gene knockdown in zebrafish particularly before the advent of other generic and targeted tools for gene manipulation [114,119,120,121]. With the increased popularity of gene editing tools such as the clustered regularly interspaced short palindromic repeats (CRIPSR)/Cas system, methods to cage guide RNA (gRNA) have also been developed, enabling precision control of gene editing and transcription [122,123,124,125,126]. Reversely, gRNAs can also be modified to deactivate CRISPR/Cas by light. For example, a recently developed CRISPRoff method [127] incorporates photocleavable *o*-nitrobenzyl groups to gRNAs which can then go through photo-induced degradation and inactivate the editing machinery. Photo-induced uncaging of a single nucleotide in the CRISPR RNA (crRNA) with subsequent release of truncated crRNAs with 15 or fewer nucleotides of target complementarity has also been shown to be sufficient to enable light-induced deactivation [128].

#### 2.2.3. Caged Peptides and Proteins

Peptides and proteins are important players in many cellular processes and are critical in maintaining homeostasis and regulating signaling pathways. Although the activities and expression of peptides and proteins can be optically manipulated by caged small molecules and caged oligonucleotides as described above, both methods only allow for indirect control of their protein targets with concomitant uncertainty in both space and time. For example, transcription and translation can take many minutes while uncaged small molecules can diffuse and affect cellular compartments that are not directly light activated. In contrast, photo-induced uncaging of peptides and proteins is instantaneous, and its effect is largely limited to a confined space set by the light. Such direct and immediate control is especially pivotal for studying biological processes that rely on fast dynamics.

Caged polypeptides and proteins typically rely on the photo-induced cleavage of C-X (i.e., C-O, C-N and C-S, but not C-C) bonds. Therefore, the amino acid residues that are usually caged are limited to those with polar or charged side chains such as cysteine (Cys), lysine (Lys), serine (Ser), tyrosine (Tyr), glutamate (Glu), glutamine (Gln), aspartate (Asp) and asparagine (Asn) [62]. Other non-polar residues such as glycine (Gly) can be caged only at the C- or N- terminus using a terminal backbone caging strategy [129]. Introducing photocaging groups to polypeptides has been accomplished in both solid-phase and solution-phase synthesis. The earliest report of a photocaged peptide describes the addition of an *o*-nitrobenzyl group to L-leucyl-L-leucine methyl ester [130], a small peptide-based lysosomal damaging agent that induces apoptosis in mast cells. Longer peptides were later successfully photocaged, targeting many amino acid residues including Cys, Ser, Lys, Glu and Asp [131,132,133,134]. Although proteins are chemically similar to peptides, caging proteins through synthetic chemistry has turned out to be much more challenging. Proteins usually possess complex structures and modifications of their amino acids could render the perturbed protein inactive or unstable. Proteins also generally require aqueous conditions, further limiting the chemical techniques that can be used for photolabile caging. Nonetheless, a few examples of synthetically caged proteins have been reported. A photoactivatable hen egg lysozyme was made by total chemical synthesis [135] and site-selective caging of whole proteins has also been achieved [136,137,138,139].

Compared to synthetic chemistry, a more powerful and established method to cage peptides and proteins has been engineered through genetic code expansion that allows incorporation of unnatural amino acids (UAAs) [140,141,142,143,144]. Genetic encoding of UAAs is made possible with installation of orthogonal transfer RNA (tRNA) synthetases and their cognate tRNAs incorporating UAAs in response to the one of three stop codons (UAG, UAA and UGA). To minimize interference with the endogenous translational machinery, a stop codon that is used less frequently by endogenous tRNA synthetase/tRNA pair is preferred. As the amber codon (UAG) is the least used in *Escherichia coli* (7–8%), the organism in which most proteins are expressed and purified in the laboratory, it is the most preferred one for the incorporation of UAAs [144]. The use of the ochre (UAA) and opal (UGA) stop codon is less common but has also been implemented [145]. Based on the genetic code expansion technology, caged amino acids critical for protein functions could be inserted in a protein of interest with little change to its overall structure. Light irradiation releases the caging groups from corresponding amino acids and restores the activity of the protein. The first photocaged protein implementing genetic incorporation was the muscle nicotinic acetylcholine receptor in which the tyrosine residue was replaced with *o*-nitrobenzyl tyrosine [146]. That study also suggested that the receptor could respond differently in *Xenopus* oocytes when the tyrosine was caged at different positions, stressing the importance of rational caging design and downstream validation. Encouragingly, the genetic incorporation of photolabile UAAs to proteins has proven widely useful in the past decades, and photoswitchable proteins have also been engineered with azobenzene-based caging groups [147]. This unique opto-chemical tool for controlling proteins has been extensively employed in the study of numerous biological processes [148,149,150] including control of ion channels [146,151,152,153], kinase and phosphatase activities [154,155,156], gene expression [157,158], epigenetic regulation [159,160], DNA recombination [161], protein localization [162,163], protein self-organization [164], intein splicing [165,166], and O-linked-N-acetylglucosaminylation [167].

**Table 1 molecules-27-06231-t001:** Some common caging groups used in photocaging. Leaving groups (X) are colored red.

Caging Group	Chemical Structure	Typical Activation Wavelength (nm)	Examples and References
2-Nitrobenzyl derivatives	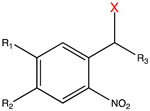	250–450	ATP [65], glutamate [72], retinoic acid [81], rapamycin [84,168], doxycycline [86], Cyc [89], Ca^2+^ [78], auxin [96], PROTAC [97], siRNA [107], MO [114,118], gRNA [122], Lys [135], Ser [135], Cys [151], Tyr [161], Gln [169], Glu [170], Ans [171], small-molecule inhibitor [172].
Coumarin derivatives	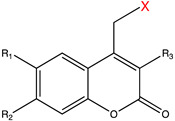	340–470	Glutamate [73], GABA [77], retinoic acid [81], tamoxifen [88], Cyc [89], PROTAC [98], mRNA [104], MO [118], Cys [173], Lys [174], Gly [175], small-molecule inhibitor [176].
7-Nitroindolinyl derivatives	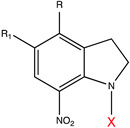	300–450	GABA [76], Asp [133], Glu [133], acetic acid [177], Ca^2+^ [178], auxin [179].
BODIPY derivatives	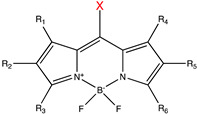	500–650	Histamine [180], dopamine [180], acetic acid [181], alcohols [182], small-molecule inhibitor [183].

## 3. Applications of Opto-Chemical Tools in Zebrafish

The technologies for the opto-chemical control of biological processes have substantially advanced in recent years. Although their working principles can be demonstrated in vitro using bacteria and cultured cells, these opto-chemical tools need to be evaluated and validated in vivo with increasing complexity such as vertebrate models. Rodents are popular vertebrate models and have been successfully used in optogenetics studies [15,184], but they often require invasive techniques for light delivery such as implantable optical fibers. With respect to opto-chemical control, efficient and uniform administration of exogenous chemical molecules in rodents is a major challenge. In contrast, the zebrafish has proven to be an ideal vertebrate model for the applications of opto-chemical tools. Firstly, zebrafish embryos are inherently transparent, and translucent adult zebrafish has also been genetically engineered via pigmentation mutation [185]. As a result, optical manipulation and imaging in zebrafish are quite convenient. Secondly, the delivery of exogenous molecules in zebrafish is also feasible through microinjection (for large molecules) or water incubation (for small molecules). Furthermore, recent advances in multiphoton microscopy [186,187,188] enable deep-tissue perturbation and imaging at the single cell level, making zebrafish an even more attractive model for broad physiological studies. In various proof of concept studies, zebrafish have been used to evaluate novel opto-chemical tools in live animals. For example, zebrafish were first used to demonstrate that protein activity can be photo-controlled by caged Cyc in vivo, at the global and the single cell levels [89]. In this study, GFP and mCherry were fused to the modified estrogen receptor and a nuclear localization signal. Upon light illumination of zebrafish embryos treated with caged Cyc, the fluorescent proteins were shown to translocate to the nucleus. Similarly, photoactivatable morpholinos targeting important genes in development were initially tested in zebrafish to evaluate the feasibility of caging strategies in vivo [114,115]. The list of opto-chemical applications in zebrafish is long and continues to grow dramatically, including modulation of neural activity, cell signaling and development, gene expression, gene editing, fluorescent imaging, and injury and regeneration (Figure 3). In the following sections, we describe representative examples of these studies to highlight the great potential and promise of applying opto-chemical tools in the zebrafish model (Table 2).

### 3.1. Modulation of Neural Activity

Neuroscience is arguably the field that has been most transformed and advanced by optogenetic and opto-chemical tools. One key advantage of optical tools is that they are able to noninvasively mimic the fast dynamics of naturally occurring neural signals, in both temporal and spatial patterns. This is especially important for neuropharmacology, where conventional drug delivery is relatively slow and imprecise in controling neural signaling. Zebrafish at the embryonic and early life stages are an ideal model for opto-chemical applications as most photochemical compounds can diffuse into embryonic tissues when added in water. For example, a rhodamine-based light sensitive molecule, optovin, was identified from a behavior-based screening assay using zebrafish [189]. Optovin can be reversibly photoactivated by violet light and binds to the transient receptor potential A1 (TRPA1) channel, resulting in alteration of motor activity in zebrafish at both early life and adult stages. Small-molecule photoswitches with azobenzene groups for glutamate receptors have also been developed and are able to drive light-dependent motility behavior in zebrafish [190,191]. A similar strategy has also been applied to the first small-molecule agonist of G-protein coupled inwardly rectifying potassium (GIRK) channels [192]. The azobenzene-based photoswitch activates GIRK channels in the dark and can be deactivated and reactivated by UV and blue light irradiation, respectively. Consequently, zebrafish treated with the photoswitchable compound displayed light-dependent swimming mobility. Rather than applying them separately, photoswitchable ligands could also be tethered to their acting neuronal receptors. Typically, genetically modified receptors are engineered to facilitate the tethering of photoswitchable ligands [193]. One benefit of this tethering method is that it allows much tighter control of the receptor activity because ligands are covalently attached to receptors, eliminating the diffusion-related concern. On the other hand, the efficient implementation of photoswitch-tethered receptors in animals still poses great challenges as the system requires delivery of both genetically engineered receptors and chemically engineered ligands. Nevertheless, the successful utility of this opto-chemical tool in vivo has been first demonstrated in zebrafish [194]. In this work, a transgenic zebrafish line was initially established to express an engineered light-gated ionotropic glutamate receptor (LiGluR), and zebrafish at early life stage were briefly (30 min) incubated with the chemically engineered ligand that could covalently attach to LiGluR. The manipulated zebrafish then exhibited light-dependent motor and visual behavior. Later studies using similar techniques to make light controllable glutamate receptors have also been implemented in zebrafish [195,196,197,198], revealing detailed neural networks that regulate development, memory, and disease.

### 3.2. Cell Signaling and Development

The zebrafish is an important vertebrate model for the study of developmental biology and cell signaling pathways. For example, the zebrafish is one of the most commonly used models in forward genetic screening to identify genes responsible for mutant phenotypes. Large-scale phenotypic screening after mutagenesis has been performed owing to the high fertility and fecundity of zebrafish, revealing thousands of mutations involved in embryonic development [199,200]. In another excellent example, zebrafish have been extensively used to study vertebrate somitogenesis which requires spatiotemporal integration of multiple signaling systems such as the fibroblast growth factor (FGF), transforming growth factor β (TGF-β), pluripotency factors (e.g., VENTX/NANOG), and Notch and Wnt signaling during anteroposterior axis elongation, cell fate choice and tissue morphogenesis [201,202,203]. Evidently, cellular signaling pathways are highly coordinated during development, acting precisely at specific times and in specific regions in the embryos. In principle, opto-chemical approaches can perturb and mimic endogenous signaling pathways with comparable spatiotemporal resolution, expanding the toolbox for precisely studying dynamic cell signaling in development. Ca^2+^ signaling plays a crucial role in various physiological processes including embryogenesis. Targeted laser-release of caged Ca^2+^ in the presomitic mesoderm (PSM) cells in zebrafish embryo shortened the mediolateral width of somites but had no significant effect on the anteroposterior length of the somites [204], suggesting a possible ubiquitous Ca^2+^ signaling in the PSM coordinating cell movements during somite segmentation. Small molecule metabolites and ligands can also be photo-controlled to regulate cell signaling and development. *All-trans* retinoic acid has been successfully caged to optically control zebrafish eye development [81], and photo-isomerization of *13-cis* to *all-trans* retinoic acid has been demonstrated to optically control hindbrain formation [205] and eye growth [206] in zebrafish embryos. In other examples, caged rapamycin has been used in zebrafish to optically control mTOR signaling, and a caged rearranged during transfection (RET) tyrosine kinase inhibitor has been developed to modulate motoneuron development in zebrafish [172]. In addition to photo-control of small molecules, larger molecules such as oligonucleotides and proteins can also be caged to optically regulate cell signaling in zebrafish development. For instance, photo-induced silencing of T-box transcription factors no tail a (ntla) and spadetail (tbx16) by cMOs causes distinct developmental defects in zebrafish embryos [114,115,207] and spatiotemporal dissection of the T-box genes with cMOs sheds light on their mechanisms by promoting medial floor plate formation and paraxial mesoderm development [207,208,209]. More recently, genetic encoding of caged unnatural amino acids has been successfully introduced in zebrafish to optically modulate in vivo cell signaling such as the MEK/ERK pathway [210].

### 3.3. Gene Expression

Optical-chemical tools have been extensively engineered to control gene expression at both transcriptional and translational levels. Many types of transcriptional machinery rely on the presence of small-molecule inducers such as estradiol, ecdysone, tamoxifen, doxycycline, rapamycin and isopropyl β-D-1-thiogalactopyranoside. Caged forms of these inducers have been developed to optically control gene expression [86,87,211,212,213], and many of them have been successfully applied in zebrafish. For example, the tamoxifen-activated Cre-ERT2/loxP system is one of the most widely used techniques to induce constitutive gene expression. The ERT2 is a genetically modified estrogen-receptor binding domain. When fused to the Cre recombinase, it forms a complex with heat shock proteins (hsp) sequestering the recombinase in the cytoplasm. Upon binding to tamoxifen, the protein fusion dissociates from its complex with chaperon proteins allowing Cre to diffuse in the nucleus to catalyze site-specific recombination, putting the gene of interest under an appropriate (e.g., tissue specific or ubiquitous) promoter. Caged tamoxifen has been synthesized to photo-induce gene expression, but it is susceptible to photoisomerization and photodegradation [89]. To address this issue, caged Cyc (Figure 4A,B) has been developed and exhibits robust photocontrol in both cultured cells and in the zebrafish embryos [89]. Besides inducing permanent genetic recombination that results in constitutive gene expression, caged Cyc can also be used to induce transient gene expression. For example, one study investigated tumor initiation by optically activating caged Cyc to induce both transient and constitutive expression of a human oncogene *kRASG12V* in zebrafish, relying on the Gal4-ERT2/UAS and Cre-ERT2/loxP systems, respectively [214] (Figure 4C,D). Photoactivable Cre has also been engineered through genetic code expansion and shown to photo-induce the expression of GFP and mCherry in zebrafish embryos [215]. On the translational level, many types of cMOs have been engineered to optically disrupt mRNA translation in zebrafish, generating valuable disease models and improving our understanding of important developmental signaling pathways [112,114,115,116,117,118,207,208,209,216].

### 3.4. Gene Editing

The discovery and optimization of the CRISPR/Cas system have profoundly revolutionized modern biological research, providing a powerful tool to genetically reprogram cellular circuits with unprecedented efficiency and accuracy [217,218,219,220]. Nevertheless, the CRISPR/Cas system still lacks precise control at the spatio-temporal scale and dose-response in complex biological systems, largely due to its genotoxicity and off-target activity. The demand for precise spatiotemporal control of CRISPR/Cas activity is even more critical for its potential clinical applications as off-target gene editing in patients can result in severe consequences. Thus, spatiotemporal control of CRISPR/Cas editing has been achieved through various strategies including small-molecule and optical activation [122,123,124,221,222,223,224,225]. Opto-chemical tools to control the CRISPR/Cas system have been engineered through several caging techniques. For example, a photocleavable single-strand DNA oligonucleotide termed a “protector” that couples to gRNA was initially designed [224]. Light irradiation cleaves the “protector” and subsequently releases the gRNA. Several types of caged gRNAs have been also developed by multiple research groups relying on different caging strategies [122,123,126,226,227]. Furthermore, the Cas protein itself has been caged through genetic code expansion [228]. In this case, several lysine residues critical for Cas activity are photocaged via site-specific incorporation of unnatural amino acids. Finally, a fusion of Cas9 with the ERT2 receptor has been engineered to photo-induce its activity transiently via uncaging of caged Cyc (B.Ducos et al., manuscript in preparation). Zebrafish serve as an excellent model for evaluating gene-editing tools in vivo due to the convenience of microinjection and genotyping. Indeed, as discussed above, many opto-chemical editing tools have been used in zebrafish, validating their utility and efficacy in vivo [122,124,126].

### 3.5. Fluorescent Imaging

The exceptional optical transparency of the zebrafish makes it a great model organism for in vivo fluorescent imaging [229]. The technological development of both fluorescent probes and microscopic instruments in the past decades has been fast and remarkable. A variety of photoactivatable fluorophores have been discovered or engineered as powerful imaging probes [82]. These probes consist of both small synthetic fluorophores including fluorescein, rhodamine, coumarin derivatives and photoactivatable fluorescent proteins such as Kaede, Dendra2, mEos2 and Dronpa. Due to their fluorescent convertibility, photoactivable fluorophores are valuable tools for cell labeling and tracking in live zebrafish. For example, Kaede can be optically converted to visualize and trace cell movements during zebrafish neuronal shaping [230] and eye morphogenesis [231]. In another study, the caged fluorescein-conjugated dextran was co-activated with cMO to label and track spatially irradiated cells in live zebrafish embryos [208]. Caged fluorescein and other photoactivatable fluorophores such as Dronpa, Dendra2 and rhodamine have also been used in zebrafish for cell lineage tracking and structure labeling [230,232,233,234,235,236,237,238,239,240,241]. In recent decades, one of the most groundbreaking techniques in light microscopy has been super-resolution imaging which allows optical resolution beyond the diffraction limit of light (~250 nm) [242,243,244]. Therefore, super-resolution imaging can visualize biological structures and processes with unprecedented details that otherwise could not be observed by conventional imaging tools. Several methods have been invented for super-resolution imaging. For example, stimulated emission depletion (STED) and reversible saturable optical linear fluorescence transitions (RESOLFT) rely on spatially patterned illumination while stochastic optical reconstruction microscopy (STORM) and photoactivated localization microscopy (PALM) stochastically turn on individual molecules within the diffraction-limited volume at different time points. Since its conception, super-resolution microscopy has been successfully applied to reveal the dynamics of subcellular structures in live organisms [148,242,245]. Opto-chemically, photoactivatable fluorescent proteins and dyes are important for super-resolution techniques such as STORM and PALM which are based on the switching and localization of single molecules. Many photoactivatable florescent proteins and synthetic dyes have been widely used for super-resolution imaging including Dendra2, mEos2, Dronpa, and rhodamine-derived dyes [246,247]. Super-resolution imaging using these photoactivable fluorophores has also been applied in various zebrafish studies [248,249,250,251].

### 3.6. Injury and Regeneration

The zebrafish is one of the most popular vertebrate models to study injury and regeneration as the animal possesses the capability to regenerate many tissues and organs including heart, spinal cord, brain, retina, hair cells, liver, kidney, pancreas, and caudal fin [252,253]. The use of optogenetic and opto-chemical tools to study zebrafish tissue injury and regeneration is emerging and has been gradually appreciated. The major advantage of optical approaches is their versatility and higher precision in specific tissue targeting. Such tools are particularly valuable when tissue-specific promoters driving the expression of tissue-damaging proteins are not available. For example, a recent study optically expressed the cytotoxic ion channel variant M2 in the zebrafish using a modified Gal4/UAS system to induce targeted cell ablation in the central nervous system and axial mesoderm [254]. In another zebrafish neuronal repair model, a photoactivatable adenylyl cyclase was used to optically stimulate neuronal regeneration by increasing cAMP levels after neuronal severing [255]. Light-induced oligomerization of the trans-activation response element DNA-binding protein 43 (TDP-43) has been recently demonstrated to model human amyotrophic lateral sclerosis in the zebrafish [256]. More recently, azobenzene-based photoswitchable ligands that targets the β1-adrenoceptor have been developed to optically disrupt cardiac rhythm in the zebrafish [257], providing a new tool for the study of cardiac physiology and pathology.

**Table 2 molecules-27-06231-t002:** Representative applications of opto-chemical tools in the zebrafish.

Study	Photocontrol Modules	Biological Targets
Modulation of neural activity	Rhodamine-based photoswitch [189], azobenzene-derived photoswitches [190,191,192,194,258]	TRPA1 ligand [189], glutamate receptor modulators [190,191,194,258], GIRK ligand [192]
Cell signaling and development	2-Nitrobenzyl-based caging [81,168,172,208], coumarin-based caging [81,210], *cis-trans* isomerization [205].	Caged Ca^2+^ [204], caged retinoic acid [81], *13-cis* retinoic acid [205], caged rapamycin [168], caged RET kinase inhibitor [172], cMOs [114,115,207,208,209], caged amino acids [210]
Gene expression	2-Nitrobenzyl-based caging [89,90,114,115,214], coumarin-based caging [89,90,215]	Caged Cyc [89,90,214], caged amino acids [215], cMOs [114,116,118,207,208,209].
Gene editing	6-Nitropiperonyloxymethylene-based caging [122,124].	Caged gRNAs [122,124,126].
Fluorescent imaging	2-Nitrobenzy-based caging [208,232,236,259], 2,3-dimethyl 2,3-dinitrobutane-based caging [234], *O*^6^-benzylguanine-based caging [237].	Caged fluorescein [208,232,234,236,237,259], Kaede [205,230,231,260], mEos [214,261,262], Dronpa [230,238], Dendra2 [239,240,241], rhodamine [233,235].
Injury and regeneration	Azobenzene-derived photoswitches [257].	β_1_-adrenoceptor ligands [257].

## 4. Challenges and Future Directions of Opto-Chemical Tools in Zebrafish

Opto-chemistry is a powerful tool and applying it in the zebrafish model holds great promise for biomedical research studies. Nonetheless, opto-chemistry is still an emerging technology with plenty of room for further optimization and advancement. Certain limitations and factors must be taken into consideration to decide whether and which opto-chemical tools are best fit for a particular study. One key subject for all opto-chemical tools is their photosensitivity. Ideally, photosensitive molecules should respond only to the inducing light, but in practice, photosensitive molecules can become active over time even in the absence of the inducing light. Such observed phenomenon as “leaky activity” can result from either unintended activation by ambient light or light-independent activation such as thermal conversion. Therefore, it is important to properly design and apply opto-chemical tools specifically tailored to the question being investigated. Critically, a non-light induced control should always be included to evaluate the leakiness and rigorously interpret the data. Another necessary control is the light-illuminated only condition (i.e., without introduction of the light sensitive molecule) as light, particularly UV and blue light and the heat generated may cause cellular changes such as DNA damage and altered neuronal activity [263,264]. Several factors should be considered to avoid or reduce unintended light activation as well as light-associated side activities. For example, light-sensitive samples should be carefully handled and kept away from ambient light during experiments. Zebrafish at embryonic and early life stages can be incubated in the dark for periods from hours to several days with minimal effects on normal development. However, it is not recommended to keep zebrafish under constant darkness after 5 days post hatching (dph) as a light-dark cycle is necessary for normal development at a later stage [265]. Adult fish can tolerate darkness much longer, but constant darkness could also cause certain behavioral and physiological changes [266] which should be taken into consideration when designing an experiment. Furthermore, the wavelength and intensity of the inducing light should be properly calibrated to achieve the best activation dynamics with no noticeable side-effects what could potentially impede data interpretation. To minimize UV and blue light induced toxicity, opto-chemical tools that rely on induction with light of longer wavelengths such as red or near-infrared light have been successfully developed [41,118,267,268]. In addition, light associated toxicity can be substantially reduced with multiphoton microscopy that use longer wavelength lasers to excite photosensitive molecules with better localization and without out-of-focus absorption [187,188,269,270].

Another common issue with photo-chemical uncaging is the potential loss of spatial control caused by diffusion resulting from either nonspecific uncaging via light scattering or intercellular diffusion of uncaged molecules. This diffusion problem could potentially limit the spatial resolution of opto-chemical tools. This is particularly the case for opto-chemical approaches using caged small molecules that can freely diffuse across cells. Several strategies can help alleviate the diffusion problem. For example, some photo-switchable small ligands can be covalently tethered to their responding receptors [193,194,196,197], preventing the small ligands from diffusing before and after photo-activation. Multi-photon excitation of caged molecules can also considerably reduce diffusion, as demonstrated in the photo-activation of caged small molecules at the single cell level in zebrafish embryos [89,90,214]. A variety of synthetic probes can selectively engage subcellular regions such as the cellular membrane [271] and mitochondria [272,273], and similar approaches can also be used to synthesize caged molecules with minimal cellular diffusion. Additionally, caged molecules can be synthetically designed with selective enzyme-substrate pairs [274] or altered charge upon photo-uncaging to improve their cellular retention [275]. Diffusion issues are less of a concern with genetically encoded light-responsive proteins, but the amino acids that can be photocaged are presently limited to only a few types including Lys, Tyr and Cys, leaving room for further expansion of the tool.

To make opto-chemical tools more accessible and valuable, it is essential to establish close collaborations among chemists, biologists, and biophysicists. First, biologists are better equipped to identify important and challenging biological questions that could be specifically addressed using opto-chemical tools. Bearing these critical needs in mind, chemists must then strive to develop opto-chemical tools that can be made readily available to biologists who typically do not have the required expertise in synthetic chemistry. Commercializing opto-chemical tools can greatly facilitate their broader application as exemplified by the wide adoption of caged neurotransmitters in neuroscience research. Other caged compounds such as caged ATP, caged fluorescein and caged Cyc (sold as Actiflash) are also commercially available. If commercialization is difficult, there should be ways for convenient sharing of these reagents. Next, the opto-chemical tools should be rigorously evaluated, and new knowledge must continue to flow between biologists and chemists to reidentify and revalidate questions and needs. For example, in the early 2000s, MOs were widely used in zebrafish and other organisms as a powerful tool for gene knockdown [111,113]. The tool soon inspired the development of photo-activatable cMOs which have been successfully applied for precise spatiotemporal gene silencing in live organisms [116,117,118,216]. However, concerns later arose as MO-induced phenotypes were often found to be different, and usually more severe than those of the corresponding mutants [276]. The potential off-target effects of MOs and cMOs subsequently required more cautious use of these reagents as well as more stringent validation of the experiments [277], and new gene manipulation tools such as CRISPR/Cas9 and CRISPR interference (CRISPRi) have become attractive for opto-chemical engineering. Lastly, the application of opto-chemical tools relies on the availability of light sources of appropriate wavelength and intensity. While UV lamps, epifluorescent and confocal microscopes can be used as common light sources, certain advanced applications using opto-chemical tools require much more complex systems which may not be accessible to most biologists. For example, multiphoton microscopy is typically required to achieve photocontrol with single-cell resolution and single cell tracking software may be required for certain applications following photo-activation. Biophysicists can contribute to advancing the technology by developing more user-friendly optical set-ups and providing relevant workshops and training to more users.

## 5. Conclusions

Opto-chemistry is still at a rapidly advancing stage, but it holds extraordinary potential for biological and biomedical research. In this review, we have described the development of various opto-chemical tools and highlighted their applications in specific zebrafish studies. While zebrafish is an excellent animal model for the use of opto-chemical tools, these tools have also been widely applied in many other model organisms such as yeast, fruit fly, worm, frog, and rodent animals. Beyond basic science research, opto-chemical tools also offer translational values in the clinical setting. For example, photo-activation of combretastatin A-4, a clinical drug for cancer treatment, has been recently reported [278]. Immune checkpoint inhibitors can also be opto-chemically designed to achieve targeted and controlled cancer therapy [279]. Optimization of current tools and development of new opto-chemical tools will further increase their power and expand their applications in an even broader range of studies.

## Figures and Tables

**Figure 1 molecules-27-06231-f001:**
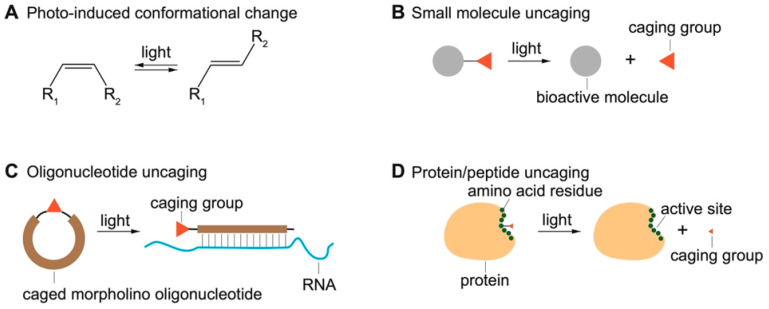
Common strategies to design opto-chemical tools. (**A**) Photo-induced conformational change which is usually reversible. Photo-induced isomerization is shown as an example. (**B**) Photo-uncaging of small molecules. (**C**) Photo-uncaging of oligonucleotides. Cyclic caged morpholino is shown as an example. (**D**) Photo-uncaging of peptides and proteins of which single amino acid residues can be caged.

**Figure 2 molecules-27-06231-f002:**
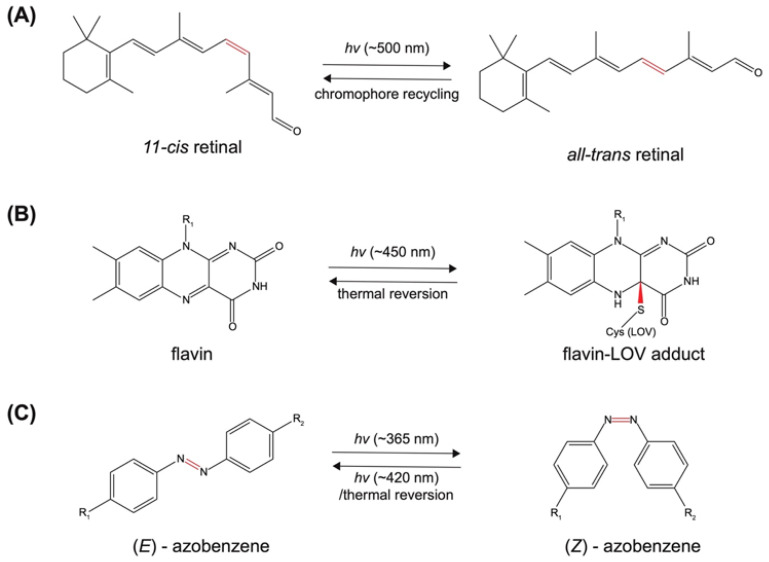
Representative examples of photoreceptor and non-photoreceptor derived photoswitches. (**A**) Isomerization of retinal in opsins. (**B**) Light-induced binding of flavin to LOV domain of naturally occurring photoreceptors. (**C**) Isomerization of azobenzene between the *Z* and *E* conformations. Light-sensitive covalent bonds are colored red.

**Figure 3 molecules-27-06231-f003:**
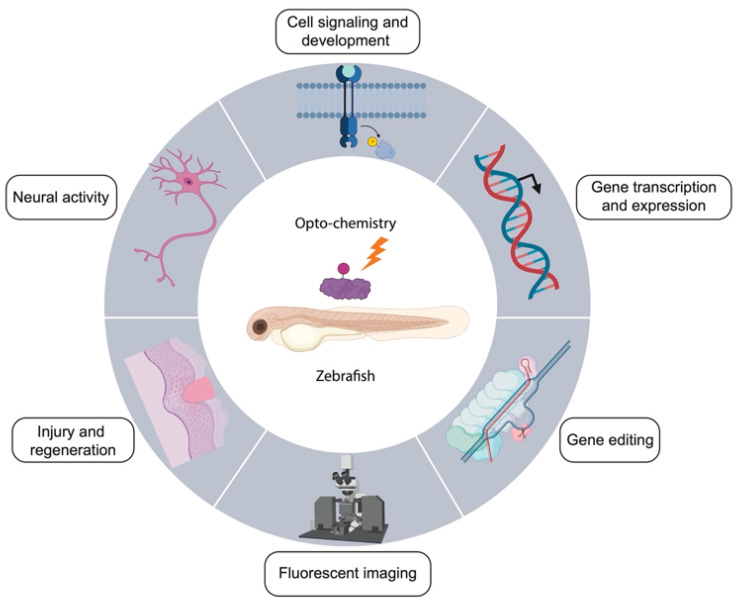
Applications of opto-chemical tools in zebrafish. The representative applications depicted are only those covered in this review. This figure is created with BioRender.com (accessed on 9 July 2022).

**Figure 4 molecules-27-06231-f004:**
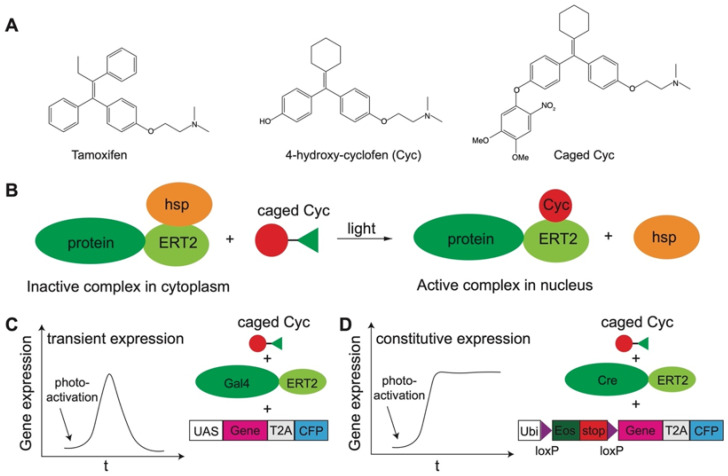
Optical control of protein activity and gene expression with caged Cyc. (**A**) Chemical structures of tamoxifen, Cyc and caged Cyc. (**B**) Optical control of protein activity with caged Cyc. Optical control of transient (**C**) or constitutive (**D**) gene expression with caged Cyc. The transient and constitutive expression of the gene of interest is driven by a UAS or a ubiquitin (Ubi) promoter, respectively. Cyan fluorescent protein (CFP) is used as an expression marker. Schemes are adapted from Refs. [82,209].

## Data Availability

Not applicable.

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
