# Peer review of "The Development and Application of Opto-Chemical Tools in the Zebrafish"

_molecules, 2022, doi:10.3390/molecules27196231_

Round 1

Reviewer 1 Report

Dear Authors,

The manuscript with title “the development and application of opto-chemical tools in the zebrafish” reviews the origin, the development and building of opto-chemical tools as well as reviewing specific examples of the application of those tools in zebrafish. In general, the manuscript is very well written, well structured and contains some informative figures and tables.

Several comments and remarks can be found below.

The introduction mentions optogenetics and opto-chemical tools. However, a clear definition of both terms is lacking. The definition of both the above terms could be contrasted with for example reporter lines. Opto-chemical tools use light to induce a change before observing and outcome, while reporter lines or immunohistochemistry with fluorescent antibodies use light to observe the outcome directly, i.e. exposure to light should not change the outcome. Continued in this paragraph, a clear definition of both reversible vs non reversible changes after illumination could be given, e.g. photoswitches vs non-reversible changes. Could the authors clarify or comment on this please.

On a semantic note, it is unclear to me which photo-chemical systems are present in nature and are harnessed by scientists to use as a tool (e.g. chromophores) and which opto-chemical systems are created by scientist to use as a tool. Thus, the definition for tool as something that is human created. A brief section on which photo-chemical systems that are present in nature and how they were harnessed by scientist (directly used or how they inspired human created tools) may help to remove confounding use of the word tool. Although most of the information is already present in section 2.1, it could be further clarified. 

Figure 2 shows photoswitches. What is elaborated on extensively in the text is the conformational change due to light (of course the topic of the review), however, the figure also indicates the reverse action (e.g. chromophore recycling, thermal revision and light recycling). Maybe explanations or examples of how reversion and recycling of these initially conformationally changed molecules are important and how that is used as part of the opto-chemical tool would be of added value to the review. I personally was missing these explanations or examples from this part of the text. The current text defines them as switches (on-off as it were) but only really discusses the “on” part. Also discussing the “off” part would complete this review further. Could the authors please add such information.

Although specific examples of applications of opto-chemical tools in zebrafish are given in section 3.5, I do miss some specific examples of use of Dendra2, mEos2 and Dronpa. Alternatively, references to examples of use of Dendra2, mEos2, Dronpa and rhodamin derived dyes could be added to table2, which are now missing and should be added for completeness in my personal view. For Kaede, only a couple examples are given, while there are many more examples. So it could be specified, not only for Kaede but also other opto-chemical tools discussed in the other sections, why specifically these examples are given. In the end a review should aim to be as inclusive as possible.

Biologically speaking, the term larva (larvae) should only be used for invertebrates. Although wildly used for a mixed range of ages in zebrafish literature, technically the use of the term larva for zebrafish is incorrect. Rather, embryo’s (0-5dpf), early life stage (6dpf-1 month), juveniles (1-3 months) and adults (+3 months) is a better terminology. The reviewer would highly appreciate for the authors to reconsider when they use the term larvae and maybe replace by the best fitting terminology. Another solution is to state ages, as this is nicely specific and avoids having to use category terms that may or may not be correct.

Smaller comments

Please check for double or lacking spaces on lines 40,107, 108, 121, 204, 232, 292, 300, 429, 506, 507, 510.

Line103: “different substituents”, these are located on the R positions of the photoswitches? Could this be clarified please?

Line 112: “are not conformationally regulated.” This statement is a little vague, so could this be clarified please?

Line 175: This section starts with a strong statement. Conventionally speaking most biological programs are initiated at the genetic level. But is this indeed true? Do we already know enough about epi-genetic (including environmental) induced programs, are they difficult to study or are they underrepresented in studies? I personally would tone down the statement “Because many biological programs are initiated at the genetic level…” or alternatively revise the statement.

Line 195: “study” should be “studies”?

Line 225: remove “s” from “takes” to form “take”.

Line 232: This sentence starts with “Therefore”, however it is not clear to me why the amino acid residues that are usually caged are limited to those with polar or charged sides. Could this be clarified please?

Line 256: suggested change “Since” to “Because”

Line 279: Although I completely understand what the authors mean by this sentence, an argument can be made about bacteria and cells not being simple systems. Especially in vitro cell culture, because when a cell is taken from its natural (bodily) environment it completely changes its gene expression, cell physiology and thus often its phenotype. This trait of in vitro cells leads often to surprising or unexpected results as well as very careful interpretations. Therefore, the statement could be revised as follows, “Although their working principles can be demonstrated in in vitro systems such as bacteria and cultured cells, these opto-chemical tools need to be evaluated and validated in vivo in biological systems with increasing complexity such as vertebrate models.”

Line 287: suggested change “transparent” to “translucent” because caspr zebrafish are not really optically transparent but rather translucent.

Line 331: suggested change “since” to “because”

Line 342: remove “later” at the start of the line.

Line 347: suggested change “mutational” to “mutant”

Line 348: suggested change “Large-scale phenotypic screening after mutagenesis…”

Line 361-362: suggested change, “mediolateral width” and “anteroposterior length”. As length of zebrafish is most often measured anteroposteriorly.

Line 444: “Other”, not clear which other are meant here. Could the authors clarify please?

Line 522: suggested change, remove “On the other hand” and start with “Several …”

Line 535: the limited number of amino acids indicated here harkens back to line 232. If clarified why polar or charged sides are important, this statement may also become clearer.

Line 570: suggested change “Review” to “review”

Line 571: suggested change, add “specific” before “zebrafish” at the end of the line.

Line 575: title separation of “glossary” is not clear.

Author Response

The manuscript with title “the development and application of opto-chemical tools in the zebrafish” reviews the origin, the development and building of opto-chemical tools as well as reviewing specific examples of the application of those tools in zebrafish. In general, the manuscript is very well written, well structured and contains some informative figures and tables.

Several comments and remarks can be found below.

Response 1: We very much thank and appreciate the reviewer’s positive feedback and comments. Responses to specific comments are listed below.

The introduction mentions optogenetics and opto-chemical tools. However, a clear definition of both terms is lacking. The definition of both the above terms could be contrasted with for example reporter lines. Opto-chemical tools use light to induce a change before observing and outcome, while reporter lines or immunohistochemistry with fluorescent antibodies use light to observe the outcome directly, i.e. exposure to light should not change the outcome. Continued in this paragraph, a clear definition of both reversible vs non reversible changes after illumination could be given, e.g. photoswitches vs non-reversible changes. Could the authors clarify or comment on this please.

Response 2: We appreciate the reviewer’s comment. The general definition of both optogenetics and opto-chemical tools implies the use of light to modulate activities of light-sensitive/activatable molecules, hence the change of cell activities. The key component here is a molecule whose activity/property can be changed by light. In the case of optogenetics, these molecules are mostly proteins that are genetically engineered and expressed. While opto-chemical tools typically rely on chemistry rather than genetical installation of light-activatable molecules. In the examples of reporter lines or fluorescent antibodies, as the reviewer already mentioned, the light interacting molecules (antibodies or fluorescent proteins) don’t change their activities/properties upon light illumination. But if they do, they can also fall into the opto-chemical category.

Regarding reversible vs non-reversible changes, reversible changes (photoswitch) means that the molecule can undergo reversible photochemistry so that many rounds of active/inactive states can be produced. On the contrary, in the non-reversible changes, molecules undergo irreversible photochemistry and the molecules remains active once uncaged so only one round of inactive/active state happens. In light of the reviewer’s comment, we have added a few sentences to clarify the definitions of optogenetics, opto-chemical tools, reversible and non-reversible changes.

On a semantic note, it is unclear to me which photo-chemical systems are present in nature and are harnessed by scientists to use as a tool (e.g. chromophores) and which opto-chemical systems are created by scientist to use as a tool. Thus, the definition for tool as something that is human created. A brief section on which photo-chemical systems that are present in nature and how they were harnessed by scientist (directly used or how they inspired human created tools) may help to remove confounding use of the word tool. Although most of the information is already present in section 2.1, it could be further clarified. 

Response 3: We apologize for the unclear description of natural photo-chemical systems/tools in this section. As the reviewer has noticed, this information is mostly presented in the following section. We considered them as tools because in most all cases, these naturally occurring molecules, even without further chemical modifications, will still need to go through proper cellular installation to use them in biological studies. For example, the LOV domain needs to be cloned and fused with proteins of interest to render light control. Other light sensitive molecules such as azobenzene are extensively redesigned to achieve various properties for specific purposes. Therefore, we think it is somehow tricky to clearly define the tools as they are.

Figure 2 shows photoswitches. What is elaborated on extensively in the text is the conformational change due to light (of course the topic of the review), however, the figure also indicates the reverse action (e.g. chromophore recycling, thermal revision and light recycling). Maybe explanations or examples of how reversion and recycling of these initially conformationally changed molecules are important and how that is used as part of the opto-chemical tool would be of added value to the review. I personally was missing these explanations or examples from this part of the text. The current text defines them as switches (on-off as it were) but only really discusses the “on” part. Also discussing the “off” part would complete this review further. Could the authors please add such information.

Response 4: We appreciate and agree with the reviewer’s comment. As mentioned in response 2, unlike the irreversible light uncaging, photoswitches allow many rounds of conversion between the inactive/active states. As an example, we have added such information and reference for cis-trans retinal conversion.

Although specific examples of applications of opto-chemical tools in zebrafish are given in section 3.5, I do miss some specific examples of use of Dendra2, mEos2 and Dronpa. Alternatively, references to examples of use of Dendra2, mEos2, Dronpa and rhodamin derived dyes could be added to table2, which are now missing and should be added for completeness in my personal view. For Kaede, only a couple examples are given, while there are many more examples. So it could be specified, not only for Kaede but also other opto-chemical tools discussed in the other sections, why specifically these examples are given. In the end a review should aim to be as inclusive as possible.

Response 5: We thank the reviewer for this comment. We have now added references for the use of Dendra2, mEos2, Dronpa and Kaeda in both main text and Table2.

Biologically speaking, the term larva (larvae) should only be used for invertebrates. Although wildly used for a mixed range of ages in zebrafish literature, technically the use of the term larva for zebrafish is incorrect. Rather, embryo’s (0-5dpf), early life stage (6dpf-1 month), juveniles (1-3 months) and adults (+3 months) is a better terminology. The reviewer would highly appreciate for the authors to reconsider when they use the term larvae and maybe replace by the best fitting terminology. Another solution is to state ages, as this is nicely specific and avoids having to use category terms that may or may not be correct.

Response 6: We appreciate and agree with the reviewer’s comment. We initially used “larvae” as it is described in the references we cited. We now have replaced “larva” with other words suggested by the reviewer whenever possible. For literature that describes “zebrafish larvae” without specifying the exact ages, we removed the “larvae” in the sentences.

Smaller comments

Please check for double or lacking spaces on lines 40,107, 108, 121, 204, 232, 292, 300, 429, 506, 507, 510.

Response 7: Corrected.

Line103: “different substituents”, these are located on the R positions of the photoswitches? Could this be clarified please?

Response 8: The substituents can be located on the phenyl ring including the R positions. We have now clarified this in the text.

Line 112: “are not conformationally regulated.” This statement is a little vague, so could this be clarified please?

Response 9: We appreciate the reviewer’s concern, and we now have clarified it as “are not conformationally regulated by light”. 

Line 175: This section starts with a strong statement. Conventionally speaking most biological programs are initiated at the genetic level. But is this indeed true? Do we already know enough about epi-genetic (including environmental) induced programs, are they difficult to study or are they underrepresented in studies? I personally would tone down the statement “Because many biological programs are initiated at the genetic level…” or alternatively revise the statement.

Response 10: We now rephrased the sentence as the reviewer has suggested.

Line 195: “study” should be “studies”?

Response 11: Corrected.

Line 225: remove “s” from “takes” to form “take”.

Response 12: Corrected.

Line 232: This sentence starts with “Therefore”, however it is not clear to me why the amino acid residues that are usually caged are limited to those with polar or charged sides. Could this be clarified please?

Response 13: As stated in the manuscript, cleavage bonds are typically C-X (C-O, C-N, C-S, but not C-C) of the amino acid side chains. Such bonds exist mostly in amino acids with polar and charged amino acids. Those with polar side chains like alanine, isoleucine, and leucine typically have C-C bonds and are difficult to be caged. We have clarified C-X bond in the text to make it more clear.

Line 256: suggested change “Since” to “Because”

Response 14: Corrected.

Line 279: Although I completely understand what the authors mean by this sentence, an argument can be made about bacteria and cells not being simple systems. Especially in vitro cell culture, because when a cell is taken from its natural (bodily) environment it completely changes its gene expression, cell physiology and thus often its phenotype. This trait of in vitro cells leads often to surprising or unexpected results as well as very careful interpretations. Therefore, the statement could be revised as follows, “Although their working principles can be demonstrated in in vitro systems such as bacteria and cultured cells, these opto-chemical tools need to be evaluated and validated in vivo in biological systems with increasing complexity such as vertebrate models.”

Response 15: We have revised the sentence as the reviewer has suggested.

Line 287: suggested change “transparent” to “translucent” because caspr zebrafish are not really optically transparent but rather translucent.

Response 16: We have replaced “transparent” with “translucent” for caspr zebrafish.

Line 331: suggested change “since” to “because”

Response 17: Corrected.

Line 342: remove “later” at the start of the line.

Response 18: Revised as suggested.

Line 347: suggested change “mutational” to “mutant”

Response 19: Revised as suggested.

Line 348: suggested change “Large-scale phenotypic screening after mutagenesis…”

Response 20: Revised as suggested.

Line 361-362: suggested change, “mediolateral width” and “anteroposterior length”. As length of zebrafish is most often measured anteroposteriorly.

Response 21: Revised as suggested.

Line 444: “Other”, not clear which other are meant here. Could the authors clarify please?

Response 22: These include other photoconvertible fluorescent proteins and dyes such as Dronpa, mEos and rhodamine dyes. We have added this information for clarification.

Line 522: suggested change, remove “On the other hand” and start with “Several …”

Response 23: Revised as suggested.

Line 535: the limited number of amino acids indicated here harkens back to line 232. If clarified why polar or charged sides are important, this statement may also become clearer.

Response 24: Please see Response 13.

Line 570: suggested change “Review” to “review”

Response 25: Revised as suggested.

Line 571: suggested change, add “specific” before “zebrafish” at the end of the line.

Response 26: Revised as suggested.

Line 575: title separation of “glossary” is not clear.

Response 27: Corrected.

Reviewer 2 Report

The review manuscript entitled The Development and Application of Opto-chemical Tools in the Zebrafish submitted by the group of authors represents a comprehensive review on the use of zebrafish, as one of the most widely adopted animal models in both basic and translational research, and the development of optochemical tools and their application based on various strategies that control biological activities with unprecedented spatiotemporal resolution.

The Introduction part should be enriched with more recent references

The figure titles are missing. Also they should be placed under the picture.

Many examples from the literature were highlighted, but the descriptive parts, like images are missing. It would be more interesting for the readers to include diagrams and images from the existing scientific publications.

Numbering of the last subchapter (future perspectives) and Conclusion chapter are wrong.

Conclusion part should be extended with more motivation to present main conclusion of the manuscript to the readers.

Author Response

The review manuscript entitled The Development and Application of Opto-chemical Tools in the Zebrafish submitted by the group of authors represents a comprehensive review on the use of zebrafish, as one of the most widely adopted animal models in both basic and translational research, and the development of optochemical tools and their application based on various strategies that control biological activities with unprecedented spatiotemporal resolution.

Response 1: We thank and appreciate the reviewer’s nice summary of this work and positive comments. Specific responses are listed below.

The Introduction part should be enriched with more recent references

Response 2: We appreciate the reviewer’s suggestion with more recent references in the Introduction. We have included some more recent references in this section. Also, we would like to state that the Introduction section here is intended for a brief intro of the review, rather than the explicit background of the topic. More background information and recent references are provided afterward in each section, and as the reviewer may have noticed, about 20% of citations (55/279) are the most recent references from 2020-2022.

The figure titles are missing. Also they should be placed under the picture.

Response 3: The figure legends were at the end of the manuscript. We now have moved it under the figures.

Many examples from the literature were highlighted, but the descriptive parts, like images are missing. It would be more interesting for the readers to include diagrams and images from the existing scientific publications.

Response 4: We thank the reviewer’s suggestion. Indeed, as the reviewer has observed, this work describes extensive examples of the development and application of opto-chemical tools. And we agree that including figures showing this information would be interesting to readers. On the other hand, we feel that it is a little improper and difficult to select particular figures from so many literatures. Instead of making a long and exhaustive list of figures, we decided to make schematic figures and tables to represent those work.

Numbering of the last subchapter (future perspectives) and Conclusion chapter are wrong.

Response 5: We noticed that the numbering errors were introduced after editorial reformatting. We have corrected the errors accordingly.

Conclusion part should be extended with more motivation to present main conclusion of the manuscript to the readers.

Response 6: We appreciate the reviewer’s comment. We have added some more paragraphs to emphasize the potential of opto-chemical tools.

Round 2

Reviewer 2 Report

All suggestions were corrected or justified briefly.